# Prediction of Wear Rate in Al/SiC Metal Matrix Composites Using a Neurosymbolic Artificial Intelligence (NSAI)-Based Algorithm

Akshansh Mishra [1],* and Vijaykumar S. Jatti [2]

1    School of Industrial and Information Engineering, Politecnico Di Milano, 20121 Milan, Italy
2    Department of Mechanical Engineering, Symbiosis Institute of Technology, Pune 412115, India; vijaykumar.jatti@sitpune.edu.in
*    Correspondence: akshansh.mishra@mail.polimi.it; Tel.: +39-35-1576-6436

**Abstract:** This research paper delves into an innovative utilization of neurosymbolic programming for forecasting wear rates in aluminum-silicon carbide (Al/SiC) metal matrix composites (MMCs). The study scrutinizes compositional transformations in MMCs with various weight percentages of SiC (0%, 3%, and 5%), employing comprehensive spectroscopic analysis. The effect of SiC integration on the compositional distribution and ratio of elements within the composite is meticulously examined. In a novel move for this field of research, the study introduces and applies neurosymbolic programming as a novel computational modeling approach. The performance of this cutting-edge methodology is compared to a traditional simple artificial neural network (ANN). The neurosymbolic algorithm exhibits superior performance, providing lower mean squared error (MSE) values and higher R-squared ($R^2$) values across both training and validation datasets. This highlights its potential for delivering more precise and resilient predictions, marking a significant development in the field. Despite the promising results, the study recognizes that the performance of the model might vary based on specific characteristics of the composite material and operational conditions. Thus, it encourages future studies to authenticate and expand these innovative findings across a wider spectrum of materials and conditions. This research represents a substantial advancement towards a more profound understanding of wear rates in Al/SiC MMCs and emphasizes the potential of the novel neurosymbolic programming in predictive modeling of complex material systems.

**Keywords:** neurosymbolic artificial intelligence; wear rate; metal matric composites; neural networks





## 1. Introduction

Aluminum and its alloys have gained significant prominence and widespread utilization across various technical fields. These versatile materials find extensive applications in industries such as aerospace, space, military, automotive, and electronics. The unique combination of desirable properties exhibited by aluminum makes it an attractive choice for diverse applications.

In the aerospace industry, aluminum and its alloys are employed in the construction of aircraft structures, including fuselages, wings, and engine components. The lightweight nature of aluminum contributes to improved fuel efficiency and enhanced performance of aircraft.

Similarly, in the space industry, aluminum plays a vital role in the construction of satellites, rockets, and spacecraft. Its lightweight properties, along with good strength and corrosion resistance, make it an ideal material for space exploration missions.

The military sector extensively utilizes aluminum and its alloys due to their strength, durability, and resistance to harsh environmental conditions. Applications range from armored vehicles and aircraft to defense equipment and weaponry components.

In the automotive industry, aluminum is widely used to manufacture engine blocks, cylinder heads, chassis components, and body panels. Its high strength-to-weight ratio helps to reduce the overall weight of vehicles, leading to improved fuel efficiency and lower emissions.

Composite materials, either engineered or naturally sourced, consist of two or more distinct materials possessing substantially different physical or chemical characteristics. These constituent elements retain their individuality at either macroscopic or microscopic scales within the final structure. The earliest instances of human-engineered composite materials were bricks formed from straw and mud for construction purposes [1–4].

Constituent materials, of which at least a percentage of each type is required, are used to make composite materials. By maintaining their relative locations, the matrix material encases and stabilizes the reinforcing materials [5–7]. The specific mechanical and physical features of the reinforcement materials help to improve the matrix's qualities.

In recent years, metal matrix composites have attracted significant attention due to their superior characteristics. Although the reinforcement often contributes many sought-after mechanical properties, these composites demonstrate anisotropic behavior and their production using traditional techniques can be challenging [8–12].

Presently, particulate-reinforced aluminum matrix composites are receiving increasing attention due to their cost-effectiveness and beneficial isotropic properties. Aluminum alloys, due to their low weight and excellent thermal conductivity, have become a favored engineering material in industries such as automotive and aviation [13–17]. These industries utilize such alloys for various high-performance components in numerous applications.

When reinforced with ceramic particles, aluminum alloys demonstrate superior mechanical properties compared to their non-reinforced counterparts, thereby making them an ideal choice for engineering applications. The production of aluminum metal matrix composites can be achieved through casting or powder metallurgy. The former method offers the benefits of lower production costs and the ability to produce larger components.

However, the casting process is not without its challenges. These include the non-wettability of ceramic particles by liquid aluminum, particle segregation, higher levels of porosity, and significant interfacial reactions due to elevated processing temperatures. The wettability of the particles can be improved through various methods, such as coating the particles with metals such as nickel and copper, introducing active elements such as magnesium into liquid aluminum, or preheating the particles prior to their integration into the liquid aluminum.

In the field of tribology, aluminum–silicon composites are of considerable importance and have garnered extensive interest for both their practical uses and fundamental attributes [18–21]. Over the past decade, these composites have seen substantial growth in their application within structural engineering, including use in the automotive, aerospace, and marine industries. The high strength-to-weight ratio of Al-Si composites makes them a favorable material choice. Furthermore, aluminum–silicon alloys form a significant group within aluminum foundry alloys.

Despite the extensive application and promising properties of aluminum–silicon composites in various industries, predicting their wear rates remains a significant challenge. This difficulty stems from the complex nature of the composites and the effects of various compositional transformations, particularly those involving the integration of silicon carbide (SiC). Existing forecasting models, such as simple artificial neural networks (ANNs), have shown some success in this area [22–25], but there is a pressing need to explore new and potentially more accurate computational modeling approaches for predicting wear rates in these Al/SiC metal matrix composites (MMCs). The previous literature has focused on the implementation of using conventional machine learning models for predicting wear rate [26–30].

The problem extends to the inherent variability in composite materials, particularly when it comes to the distribution and ratio of elements within them. This variability means

that the performance of any given model could change based on the specific characteristics of the composite material and the operational conditions under consideration. Thus, there is a need for a robust, adaptable, and accurate model to predict wear rates in these composites, which not only considers the specific material characteristics and operational conditions but also accommodates the integration of different weight percentages of SiC.

The potential application of neurosymbolic programming, as a novel computational modeling approach in this field, represents an exciting avenue for exploration. However, this approach needs to be thoroughly tested and compared with existing methods to ascertain its effectiveness and adaptability across various materials and conditions. This need for a robust and resilient predictive model for wear rates in Al/SiC MMCs forms the basis of our research problem.

This research paper presents a significant contribution by employing neurosymbolic programming to predict wear rates in aluminum–silicon carbide (Al/SiC) metal matrix composites (MMCs). What sets this study apart from similar works in the field is its innovative application of the neurosymbolic algorithm, which combines the interpretability of decision trees with the learning capabilities of neural networks. By doing so, it enables a comprehensive analysis of compositional transformations and spectroscopic analysis findings, leading to improved predictive accuracy.

The evaluation between the neurosymbolic algorithm and a conventional simple artificial neural network (ANN) is a crucial component of this research. The outcomes emphasize the superior performance of the neurosymbolic approach, which is supported by reduced mean squared error (MSE) values and greater R-squared ($R^2$) values throughout both training and validation datasets. This advantage highlights the neurosymbolic approach's ability to deliver more accurate and durable forecasts, marking a significant leap in the field.

## 2. Materials and Methods

This section presents the experimental procedure employed in the current research work. The aluminum alloy HE-30 has a high tensile strength among all Al-Mg alloys, with good corrosion resistance, good machinability and good weldability. The primary constituent in this alloy is magnesium and Table 1 gives the detail chemical composition of HE-30 Al alloy as measured via Spectro analysis.

**Table 1.** Spectro analysis of base material.

| Cr | Cu | Fe | Mg | Mn | Ni | Pb | |
|---|---|---|---|---|---|---|---|
| <0.005 | <0.005 | 0.172 | 0.441 | 0.0206 | 0.0075 | <0.05 | |
| **Si** | **Sn** | **Sr** | **Ti** | **V** | **Zn** | **Zr** | **Al** |
| 0.46 | 0.0216 | <0.01 | 0.0167 | <0.005 | 0.00558 | <0.005 | 98.8 |

In this investigation, silicon carbide was employed as a reinforcing material. Particles of silicon carbon of around 25 m in size were purchased on the open market. Tetrahedral carbon and silicon atoms are bonded tightly together to form silicon carbide, which is its chemical name. This results in a material that is exceedingly strong and hard. Up to 800 °C, no acids, alkalis, or molten salts can attack silicon carbide. This material has remarkable thermal shock resistance due to its strong thermal conductivity, minimal thermal expansion, and high strength.

Composite specimens were created using a liquid metallurgical process. The most cost-effective way to create composites with irregular or discontinuous fibers is with this technique. To maintain the matrix alloy in the semi-solid state, the temperature of the alloy is first raised to its melting point and then gradually decreased until it is below the liquidus temperature. The preheated SiC particles are introduced to the slurry and stirred at this temperature. The temperature of the composite slurry was then raised to a fully liquid

state, and an automatic stirring mechanism was then used. The hot melt was finally poured into the permanent cast iron mold.

After being cleaned to eliminate surface imperfections, an identified amount of HE-30 alloy bars were loaded into an oil-fired furnace and heated to 760 °C. In order to dry out the silicon carbide particles, they were also warmed to 800 °C. Hexa-chloro-ethane tablets were used to purge the molten aluminum alloy of any trapped gases in order to reduce porosity. The aluminum alloy was then melted and degassed. To get rid of the slag content, the slag powder was sprayed. The base metal was strengthened with warmed silicon carbide particles by weight ratio. With the aid of a stirrer, the melt was stirred. The churning was maintained for 10 min at a 400 rpm impeller speed. The melt temperature was kept at 720 °C while the silicon carbide particles were added.

The permanently heated metallic molds were filled with the molten alloy containing reinforced particles. At 700 °C, the pouring temperature was held constant. After that, the melt was allowed to set up in the molds. Three and five weight percents of silicon carbide were used to make the composite. The specimens were machined to the necessary size for wear tests after being removed from the molds that gave them their cylindrical shape.

The dry sliding wear properties of the composite specimens were assessed using DUCOM pin-on-disc sliding wear testing equipment (Manufacturer: DUCOM, Bangalore, India) as shown in Figure 1. The dry sliding wear tests were carried out in accordance with ASTM G99-95 guidelines. Acetone was used to clean the pin, and a digital electronic balance was used to determine its initial mass. Afterward, during the test, the pin was kept placed against a rotating EN-32 steel disc (counter face) with a hardness of 65 HRC. Variations in the applied normal load, sliding speed, and sliding distance were made during the testing. Following acetone cleaning, the pin's final mass was determined at the conclusion of each test. The pin's mass loss from sliding wear was determined by the difference between its original and final masses. The pin's associated density measurements were used to compute the volume loss due to wear. The composite pins' wear rate was then determined.

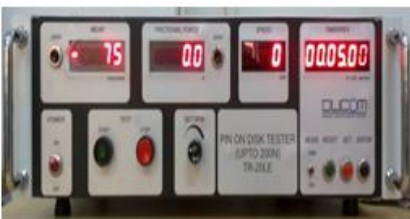
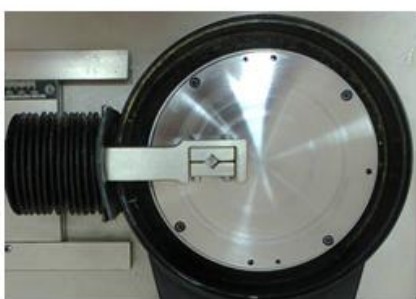
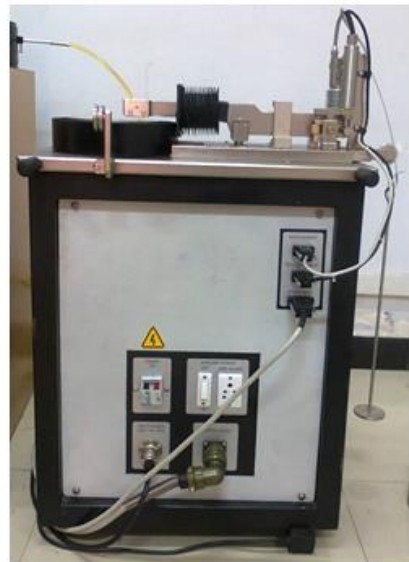

**Figure 1.** Wear test-Pin on disc.

The procedure for conducting the wear test is as follows:

Initially, the test specimen is precisely weighed using an advanced digital balance, and its initial mass is meticulously logged. Subsequently, the specimen is fastened securely using the notch, and its surface is tactically aligned to make contact with the disc. This is followed by an adjustment of the track radius in line with the test's specific needs. Once the specimen is correctly positioned, predetermined normal loads are applied, and the sliding velocity is established in accordance with the test specifications. The test is then carried

out over a calculated time interval to traverse the defined distance. Upon completion of the testing period, the worn specimen is gently detached, and its concluding mass is measured and documented. The disparity between the initial and concluding weights gives an accurate indication of the extent of wear endured by the specimen during the test. This procedure is subsequently reiterated for additional specimens that exhibit different volume percentages and may be subjected to diverse test parameters. This approach allows for the comparative analysis of wear characteristics under a range of conditions.

Table 2 displays the wear parameters that were selected for the tests in accordance with the pilot trials, literature analysis, and machine capacity. To determine the realistic limitations of the aforementioned factors in order for the wear to happen in steady state, pilot experiments were carried out following ASTM G99-95. The pin utilized for the wear test has an 8 mm diameter and a 25 mm length.

**Table 2.** Process parameters with their values.

| Level | Sliding Speed (m/s) | Load (N) | Sliding Distance (m) | Wt% Sic |
|:-----:|:-------------------:|:--------:|:--------------------:|:-------:|
| 1 | 0.314 | 9.81 | 500 | 0 |
| 2 | 0.942 | 29.43 | 1000 | 3 |
| 3 | 1.570 | 49.05 | 1500 | 5 |

Through robust experiment design, the Taguchi technique seeks to minimize variation in a process. The method's overarching goal is to generate high-quality output at a cheap cost to the maker. Dr. Genichi Taguchi of Japan created the Taguchi method, and he has continued to use that variant. Therefore, poor process quality has an impact on society as well as the producer. In order to explore how various parameters influence the mean and variance of a process performance characteristic that indicates how well the process is doing, he created a system for designing experiments.

By organizing the variables influencing the procedure and the magnitudes at which they should be shifted using orthogonal arrays, Taguchi's experimental design can collect the data required to identify the variables that have the greatest influence on product quality with the least amount of experimentation, saving time and resources. Analysis of variance was utilized to determine the important process parameters. As shown in Table 3, an L9 orthogonal array with 9 rows and 4 columns was selected for the current experiment.

**Table 3.** Orthogonal array for $L_9(3^4)$ Taguchi design for wear test.

| Standard Run | Sliding Speed (m/s) | Load (N) | Sliding Distance (m) | Wt% of SiC |
|:------------:|:-------------------:|:--------:|:--------------------:|:----------:|
| 1 | 0.314 | 9.81 | 500 | 0 |
| 2 | 0.314 | 29.43 | 1000 | 3 |
| 3 | 0.314 | 49.05 | 1500 | 5 |
| 4 | 0.942 | 9.81 | 1000 | 5 |
| 5 | 0.942 | 29.43 | 1500 | 0 |
| 6 | 0.942 | 49.05 | 500 | 3 |
| 7 | 1.570 | 9.81 | 1500 | 3 |
| 8 | 1.570 | 29.43 | 500 | 5 |
| 9 | 1.570 | 49.05 | 1000 | 0 |

In order to facilitate smooth integration on the Google Colab platform, the experimental data collected was translated into a CSV file format. Then, to process the data, the neurosymbolic programming method was used. The dataset was artificially enlarged to include a substantial 1000 data points in order to improve the model's precision.

The neural nets and symbolic AI are combined in the neurosymbolic programming method. The neural network structure consists of layers with 32 and 16 hidden units that are closely connected and end in a single output neuron. By using a rectified linear unit (ReLU) activation function, nonlinearity is incorporated into the model. The mean squared error (MSE) loss function directs the optimization process when utilizing the Adam optimizer for training. The model is trained using training and validation datasets over 2000 epochs with a 32-person batch size. The learnt features are subsequently extracted from the input data using a trained neural network model. A decision tree, built with the DecisionTreeRegressor provided by the scikit-learn module, serves as the model's symbolic representation. The decision tree's maximum depth is four in order to avoid overfitting. As a result, the trade-off among complexity of models and generalization ability is balanced.

The choice of our prediction model, neurosymbolic programming, and the associated parameters were influenced by several critical elements to ensure reliable, robust outcomes. The selection of neurosymbolic programming was driven by its distinct ability to merge symbolic reasoning with deep learning, enabling superior generalizations from limited data and yielding more interpretable predictions.

The chosen parameters were carefully identified based on a thorough spectroscopic analysis and exhaustive review of relevant literature. They were believed to significantly influence the wear rates in aluminum–silicon carbide (Al/SiC) metal matrix composites (MMCs). Aspects such as the weight percentages of SiC (0%, 3%, 5%) within the composites, and the distribution and ratio of constituent elements within the MMCs were diligently taken into account during the modeling phase.

Moreover, we used a strategic approach for parameter tuning to maximize our model's performance. Our data was partitioned into training, validation, and testing sets. The training set facilitated learning the model parameters, whereas the validation set aided in refining these parameters and selecting the optimal model configuration. The final appraisal of the model, based on mean squared error (MSE) and R-squared ($R^2$) values, was conducted using the test set. These metric features were then juxtaposed against those of a basic artificial neural network (ANN) model for comparative analysis.

For the purposes of this investigation, our dataset was meticulously divided into two interconnected but separate subsets: training and validation datasets. We utilized a significant portion, 80%, of the entire dataset to establish and instruct both the neurosymbolic programming model and the simple artificial neural network (ANN). This training dataset facilitated model learning by enabling the recognition of patterns and correlations inherent in the data.

The remaining 20% of the total data functioned as the validation (or testing) dataset, which was not revealed to the models during the training phase. This segment of the data was critical in appraising the performance of the models, specifically in gauging their predictive proficiency with respect to new, unseen data.

By segmenting the data into training and validation subsets, we ensured that our models were resistant to overfitting and retained their ability to yield accurate predictions when presented with new data. While both subsets are drawn from the same comprehensive dataset and exhibit similar characteristics, they were kept distinct to prevent the models from merely committing the training data to memory. Instead, the models were encouraged to extrapolate from learned patterns to make predictions.

It is crucial to note that, despite their similarities, the validation dataset was not used in any capacity during the training process. This practice was maintained to guarantee an impartial evaluation of the models' capabilities and their effectiveness in extrapolating to unseen data. This rigorous methodology allowed us to reliably assess the efficacy of our models, corroborating their potential for practical application in the realm of wear rate prediction in Al/SiC MMCs.

The generation of our data sets was based on nine sets of experimental data. From these data, we synthetically created 1000 data points using established scientific principles and statistical methodologies. These newly formed data points maintain the integrity of

the original experimental data while providing sufficient volume for effective machine learning processes.

We divided this expanded dataset into training and validation subsets, ensuring a robust evaluation of our model's performance. The synthetic generation of data points enabled us to build an adequately sized training set, allowing for substantial learning and pattern recognition for the models. Simultaneously, a meaningful validation set was reserved for unbiased assessment of the model's ability to generalize to unseen data.

## 3. Results

### 3.1. Spectroscopic Analysis

Spectroscopic analysis was performed to explore the proportion of various constituent materials within the composite. This investigation specifically examined samples containing 0%, 3%, and 5% weight percentages of silicon carbide. The results are shown in Table 4 and Figure 2. Upon examining the spectroscopic analysis results, several key observations can be made concerning the distribution of elements within composites containing different weight percentages of silicon carbide (SiC).

**Table 4.** % of different elements in MMC.

| Elements/Composites | 0%-SiC | 3%-SiC | 5%-SiC |
|:---:|:---:|:---:|:---:|
| Al | 98.8 | 98.1 | 97.7 |
| Cr | <0.005 | 0.0115 | 0.0118 |
| Cu | <0.005 | 0.0281 | 0.0332 |
| Fe | 0.172 | 0.272 | 0.36 |
| Mg | 0.441 | 0.394 | 0.454 |
| Mn | 0.0206 | 0.328 | 0.348 |
| Ni | 0.0075 | 0.00839 | 0.0089 |
| Pb | <0.05 | <0.05 | <0.05 |
| Si | 0.46 | 0.705 | 0.894 |
| Sn | 0.0216 | 0.0199 | 0.0209 |
| Sr | <0.01 | <0.01 | <0.01 |
| Ti | 0.0167 | <0.0201 | 0.0149 |
| V | <0.005 | 0.00924 | 0.00804 |
| Zn | 0.00558 | 0.0337 | 0.0361 |
| Zr | <0.005 | <0.005 | <0.005 |

The most abundant element within the composites across all samples was aluminum (Al), making up between 97.7% and 98.8% of the total weight. Notably, the percentage of aluminum decreased as the weight percentage of SiC increased, indicating a possible displacement effect by the added SiC.

Interestingly, the presence of silicon (Si), an element in silicon carbide, increased as the weight percentage of SiC increased, suggesting a direct correlation between SiC addition and Si content. This increase was from 0.46% at 0% SiC to 0.894% at 5% SiC.

Minor elements such as chromium (Cr), copper (Cu), iron (Fe), magnesium (Mg), manganese (Mn), nickel (Ni), and zinc (Zn) also exhibited an increasing trend with the addition of SiC, with iron and manganese showing the most significant increase. This suggests that the introduction of SiC may have some influence on the dispersion of these minor elements within the composite.

Certain elements, including lead (Pb), strontium (Sr), titanium (Ti), vanadium (V), and zirconium (Zr), were present in such small amounts that they fell below the detection limit

in some or all samples. For these elements, it was challenging to ascertain any clear trends relative to the SiC content.

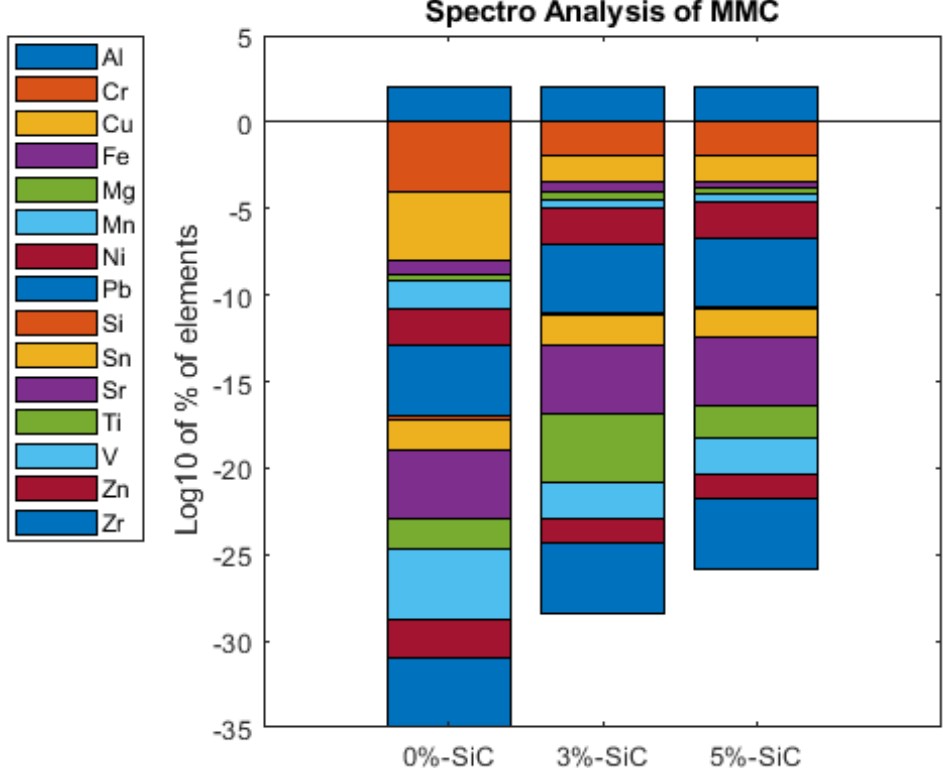

**Figure 2.** Spectroscopic analysis of the MMC in the present work.

### 3.2. Microstructural Analysis

Figure 3 displays optical micrographs of an unreinforced alloy, as well as composites with 3% and 5% volume fractions of reinforcements. These specimens' microstructure study reveals that the SiC particles are evenly dispersed throughout the matrix. Porosity was found to be present around the SiC particles, nevertheless. This observation might be explained by the way aluminum alloy wets. Additionally, it can be shown from the optical micrographs that adding more volume fractions of the particle reinforcements causes the specimens' porosity to grow.

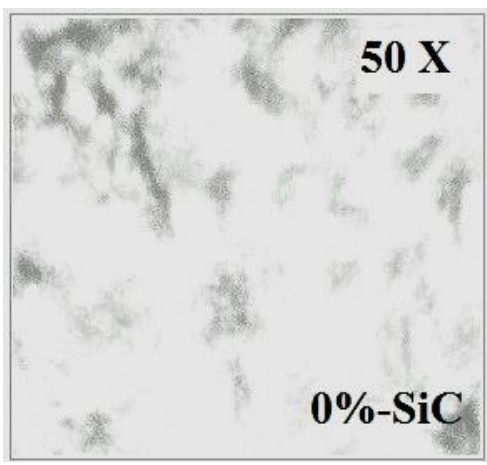

(**a**) HE-30 Alloy (0%-SiC)

**Figure 3.** *Cont.*

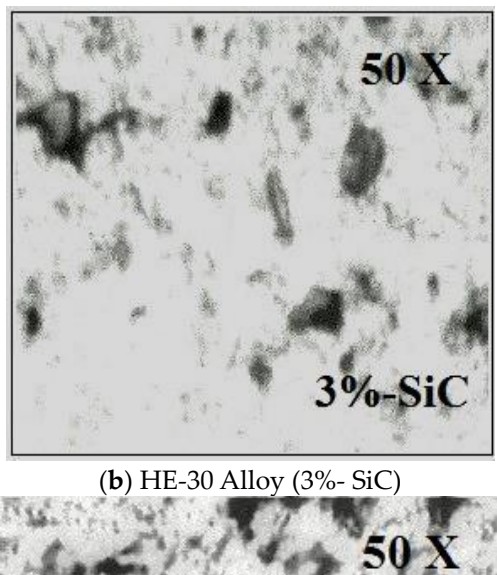

(**b**) HE-30 Alloy (3%- SiC)

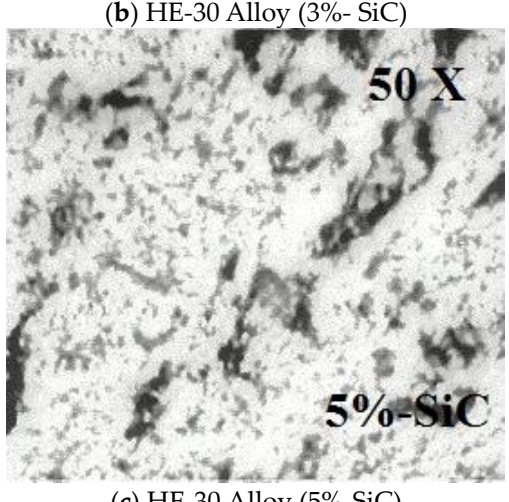

(**c**) HE-30 Alloy (5%-SiC)

**Figure 3.** Microstructure of composites.

*3.3. Wear Test Analysis*

A pin-on-disc apparatus was used to conduct the dry sliding wear test. The important parameters were identified using the signal-to-noise (S/N) ratio and analysis of variance approaches. The impact of noise elements on performance characteristics is assessed using the S/N ratios. S/N ratios assess both the degree of variation in the response data and how closely the average response comes to the target, and three of them are regarded as common and commonly used. The better of them are nominally better, higher better, and smaller better. In this study, wear rate was minimized by using the rule of smaller is better. The observed values for the $L_9$ ($3^4$) orthogonal array are shown in Table 5.

The signal-to-noise ratio (S/N ratio) measures how sensitive the quality feature under inquiry is to uncontrollable circumstances in the experiment. A higher S/N ratio is always preferred because it shows that the product will vary less from the desired value. Table 5 makes it clear that experiment number 7 has the highest S/N ratio and the smallest wear rate. The sliding speed, followed by the load, has the greatest influence on the wear rate, according to an analysis of the response table.

The most effective combination of factors and their respective amounts for producing the lowest wear rate can be identified by consulting the primary impact plots for the S/N ratio (Figure 4). The best configuration is determined to be A3B1C2D2, which corresponds to sliding speeds of 1.57 m/s at level 3, a load of 9.81 N at level 1, a sliding distance of 1000 m at level 2, and a SiC weight percentage of 5 wt% at level 2. It is anticipated that using these parameter settings will effectively reduce the wear rate.

**Table 5.** Orthogonal array of Taguchi for wear L$_9$ array [31].

| Standard Run | Sliding Speed (m/s) | Load (N) | Sliding Distance (m) | SiC wt% | Wear Rate $10^{-11}$ (m$^2$/N) | S/N Ratio |
|---|---|---|---|---|---|---|
| 1 | 0.314 | 9.81 | 500 | 0 | 9.330 | −19.39 |
| 2 | 0.314 | 29.43 | 1000 | 3 | 6.370 | −16.08 |
| 3 | 0.314 | 49.05 | 1500 | 5 | 4.598 | −13.25 |
| 4 | 0.942 | 9.81 | 1000 | 5 | 0.111 | 19.09 |
| 5 | 0.942 | 29.43 | 1500 | 0 | 2.160 | −6.68 |
| 6 | 0.942 | 49.05 | 500 | 3 | 0.100 | 20 |
| 7 | 1.570 | 9.81 | 1500 | 3 | 0.050 | 26.02 |
| 8 | 1.570 | 29.43 | 500 | 5 | 0.453 | 6.87 |
| 9 | 1.570 | 49.05 | 1000 | 0 | 0.141 | 17.01 |

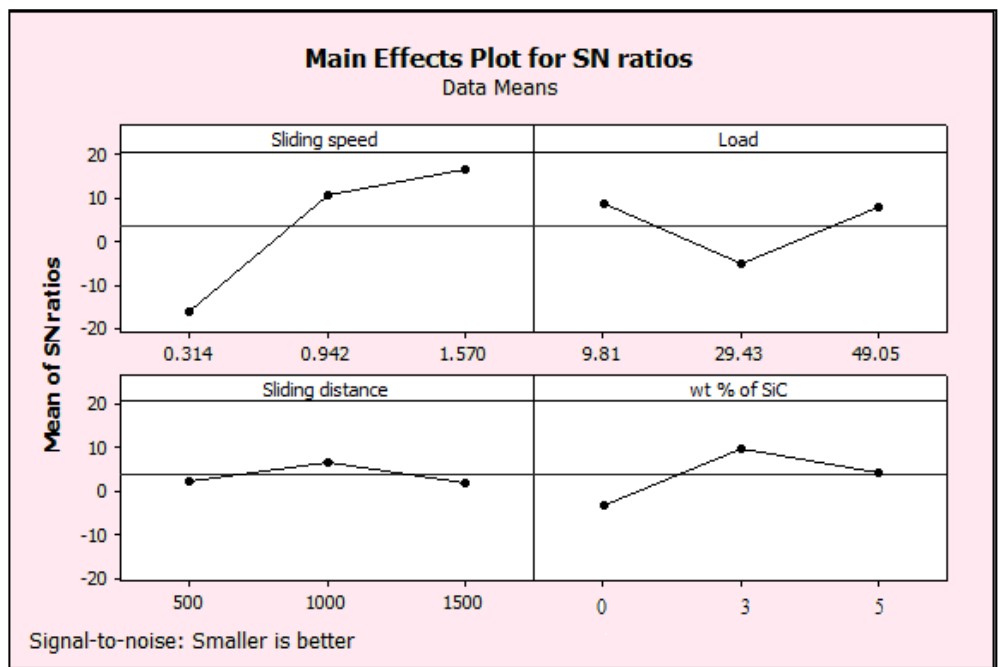

**Figure 4.** S/N ratio plots for mean effect [31].

Table 6 depicts the response table for wear rate, sliding speed, load weight % of SiC followed by sliding distance which affects the wear rate.

**Table 6.** Response table for wear rate [31].

| Level | Sliding Speed | Load | Sliding Distance | Wt.% of SiC |
|---|---|---|---|---|
| 1 | −16.244 | 8.572 | 2.493 | −3.024 |
| 2 | 10.801 | −5.298 | 6.675 | 9.979 |
| 3 | 16.638 | 7.921 | 2.027 | 4.240 |
| Delta | 32.882 | 13.87 | 4.648 | 13.003 |
| Rank | 1 | 2 | 4 | 3 |

Analysis of variance is used to determine which design factors have a significant impact on a quality feature. To do this, the contributions from each of the design parameters and the error are divided into the overall variability of the S/N ratios, which is calculated as the sum of the squared deviations from the total mean S/N ratio. First, a calculation is made to determine the total sum of squared deviations (SST) from the overall mean S/N ratio. The sum of squared deviations (SST) is then broken down into its two constituent

parts: the sum of squared deviations (SSd) resulting from every design parameter and the sum of squared error (SSe). The ratio of the sum of squared deviations (SSd) attributable to each design parameter to the total sum of squared deviations (SST) represents the percentage contribution made by each design parameter to the total sum of squared deviations. The Fisher F-test is a statistical method used to pinpoint the parameter that significantly affects a quality attribute. The mean of squared deviations (SSm) resulting from each design parameter must be computed prior to running the F-test. The sum of squared deviations (SSd) divided by the total number of degrees of freedom associated with the design parameter gives us the mean of squared deviations (SSm). The F-value is then just the ratio of the mean squared deviations (SSm) to the mean squared error for each design parameter. F generally indicates that a modification in the design parameter has a considerable impact on the quality feature if it is more than four.

Tables 7 and 8 show the result of analysis of variance for the influence of various operating parameters on the wear loss. This analysis was carried out at a level of significance of 5% (i.e., the level of confidence 95%). The result showed that the sliding speed (84.46%) has major influence on wear rate. Load (4.65%) and SiC wt% (8.31%) have moderate influence on wear rate and sliding distance (2.39%) has negligible influence on wear rate.

**Table 7.** Summary of ANOVA [31].

| Factors | Degree of Freedom | Sum of Squares | Mean Square | % Contribution |
|---|---|---|---|---|
| Sliding speed | 2 | 78.960 | 39.480 | 84.46 |
| Load | 2 | 4.341 | 2.171 | 4.65 |
| Sliding distance | 2 | 2.236 | 1.118 | 2.39 |
| SiC wt% | 2 | 7.757 | 3.879 | 8.31 |
| Error | 0 | - | - | |
| Total | 8 | 93.294 | | |

**Table 8.** Summary of pooled ANOVA [31].

| Factors | Degree of Freedom | Sum of Squares | Mean Square | F-Test | % Contribution |
|---|---|---|---|---|---|
| Sliding speed | 2 | 78.960 | 39.480 | 35.31 | 84.46 |
| Load | 2 | 4.341 | 2.171 | 1.94 | 4.65 |
| SiC wt% | 2 | 7.757 | 3.879 | 3.47 | 8.31 |
| Error | 2 | 2.236 | 1.118 | | |
| Total | 8 | 93.294 | | | |

### 3.4. Wear Rate Prediction Using Neurosymbolic Algorithm

By combining the benefits of neural networks and symbolic artificial intelligence, neurosymbolic programming creates a model that is more reliable. An initial neural network structure with three layers—an input layer, two hidden layers, and an output layer—was developed for this investigation. A single output neuron was reached after the first hidden layer, which had 32 units, and the second hidden layer, which had 16 units. The rectified linear unit (ReLU) activation function was used to provide nonlinearity to the model, as shown in Figure 5.

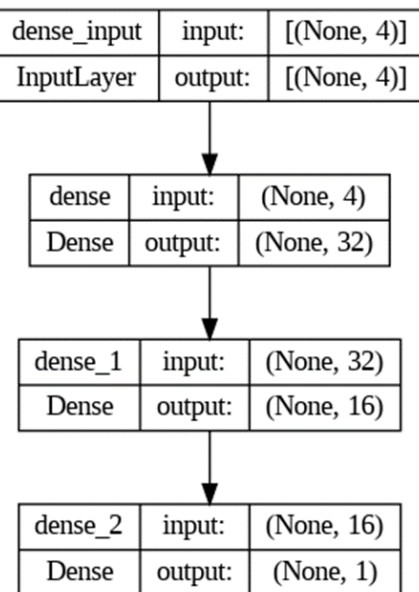

**Figure 5.** The architecture of the neural network in the current work.

Table 9 presents the calculated MSE and $R^2$ values for both the simple ANN and the neurosymbolic approach-based algorithm.

**Table 9.** Comparison of metric features of used algorithms.

| Algorithm | MSE (Train) | MSE (Validation) | $R^2$ (Train) | $R^2$ (Validation) |
|---|---|---|---|---|
| Simple ANN | 0.9901 | 1.0296 | 0.8667 | 0.8405 |
| Neurosymbolic | 0.4608 | 0.5543 | 0.9379 | 0.9141 |

## 4. Discussion

The impact of sliding speed on wear rate is presented in Table 6. The table suggests a strong correlation between the wear mechanism and sliding speed. It is evident that at lower sliding speeds, the composite exhibits higher wear rates. However, as the sliding speed increases from 0.314 to 1.570 m/s, a decrease in wear rate is observed compared to the lower sliding speeds.

Both the applied load and the volume fraction of reinforcement have an impact on the wear behavior of composite materials. The impact of load on the rate of wear for different reinforcing percentages is shown in Table 5. It is clear that the wear rate lowers for various reinforcing percentages as the applied stress rises from 9.81 to 49.05 N. Increased SiC content limits the matrix material's ability to flow and deform under stress. At lower loads, 0% reinforcement has a higher wear rate than 5% reinforcement, which has a lower wear rate for the same load. The wear rate for 3% reinforcement is also lower than for 0% and 5% reinforcement. This behavior may be attributed to the weakened bonding between SiC and the matrix material.

SiC particles, which serve as load-supporting components, are responsible for the composite materials' exceptional wear resistance. The impact of wear rate is shown in Table 5 for various combinations of speed, load, and distance. With regard to speed, load, and distance, a decrease in wear rate is seen when the percentage of reinforcement rises from 0% to 5%. The inclusion of SiC particles makes the composite harder, further increasing its resistance to wear.

The presence of particles from wear debris indicates that both adhesive and abrasive wear have occurred during the wear test, causing the plastic shearing of surface irregularities. The SEM micrographs shown in Figure 6 also reveal areas of damage, potentially

resulting in flake-shaped debris. When it comes to aluminum-based metal matrix composites (MMCs) that contain reinforcing phases such as SiC, the reinforcement material tends to act as a secondary abrasive against the opposing surface, thereby increasing wear on the counter face. Moreover, as the reinforcement material is released as wear debris, it acts as a third-body abrasive, affecting both the matrix and reinforcement surfaces. The matrix alloy's worn surface exhibits more pronounced grooves, particularly in composites with lower volume fractions of SiC reinforcement. These grooves undergo severe plastic deformation, resulting in significant wear.

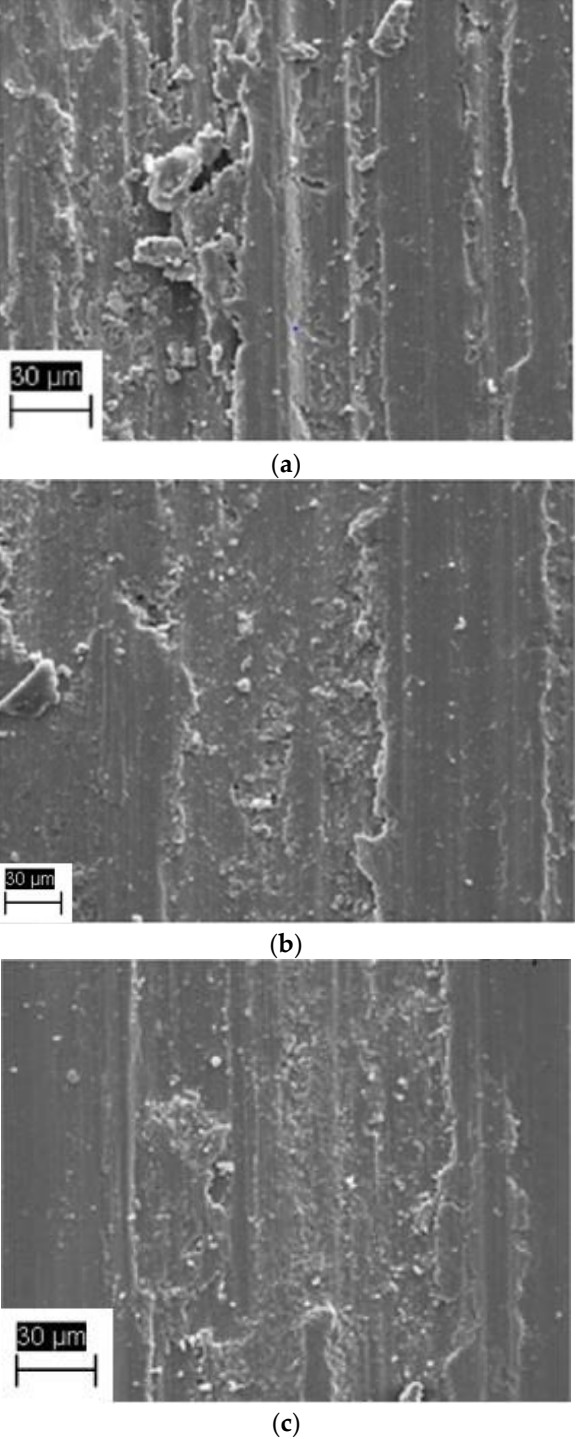

(**a**)

(**b**)

(**c**)

**Figure 6.** SEM Images of (**a**) HE 30 alloy wear scar marks at 500×, (**b**) HE 30 alloy + 3 wt% SiC wear scar marks at 500×, (**c**) HE 30 alloy + 5 wt% SiC wear scar marks at 500×.

We identified many significant findings after carefully examining the evaluation measures for both the simple artificial neural network (ANN) and the neurosymbolic algorithms.

The average squared difference between the estimated and true values, which is measured by the mean squared error (MSE), was dramatically decreased in the neurosymbolic algorithm during both the training and validation phases. The MSE values for the neurosymbolic algorithm were 0.4608 and 0.5543 for training and validation, respectively. By comparison, the simple ANN reported MSE values of 0.9901 and 1.0296, which were much higher. A lower MSE indicates a better model fit to the data, which means that the neurosymbolic algorithm will perform better.

We also employed the coefficient of determination, or R-squared ($R^2$), to determine the proportion of variance in the dependent variable that could be predicted from the independent variables. $R^2$ values are on a scale from 0 to 1, where 1 signifies an ideal fit. The neurosymbolic algorithm outperformed with $R^2$ values of 0.9379 and 0.9141 for training and validation, respectively, in comparison to the simple ANN's $R^2$ values of 0.8667 and 0.8405. The higher $R^2$ values indicate a superior predictive accuracy and model performance by the neurosymbolic algorithm.

When evaluating a machine learning model such as the neurosymbolic programming methodology, it is crucial to compare the actual and estimated values for both the training and validation datasets. This comparison allows researchers to assess the model's performance, examine its ability to generalize, and ensure that it neither overfits nor underfits the data. The actual values represent the observed target outputs in the datasets, while the estimated values are the model's predictions of the outputs based on the input features, aiming to be as accurate as possible.

Analyzing these values within the training set helps evaluate the model's capacity to identify underlying patterns and relationships during the learning process. While achieving a good fit within the training set is important, it does not guarantee the model's effectiveness. Overfitting can occur when the model excessively tailors itself to the training data, memorizing noise or capturing irrelevant correlations. Therefore, it is essential to scrutinize the model's performance within the validation set, which comprises data that the model has not encountered during training. This evaluation provides an estimation of the model's potential to generalize and make accurate predictions on unseen data.

The framework of the implemented NSAI algorithm in present work is shown in Figure 7. The implementation of the neurosymbolic algorithm in this study involves an innovative coupling of a decision tree and a neural network, aiming to improve the predictive performance on the wear rates in aluminum–silicon carbide (Al/SiC) metal matrix composites (MMCs). In this application, the symbolic component of the neurosymbolic algorithm, represented by the decision tree, is utilized to logically structure and decipher the relationships and interactions between different variables that may influence the wear rates in MMCs, such as the varying weight percentages of SiC, compositional transformations, and spectroscopic analysis findings. This approach harnesses the interpretability and transparency of decision trees, which allows for a systematic breakdown of complex relationships in the data.

The neural network component, on the other hand, serves as the subsymbolic or non-symbolic element of the algorithm, focusing on detecting and learning from the subtle, non-linear patterns in the data that are often elusive to symbolic models. In this context, the neural network processes and learns from the information and relationships identified and structured by the decision tree. It operates on compositional transformations and spectroscopic findings, improving the predictive power of the model by leveraging its capacity for high-dimensional data representation and learning from complex, non-linear relationships.

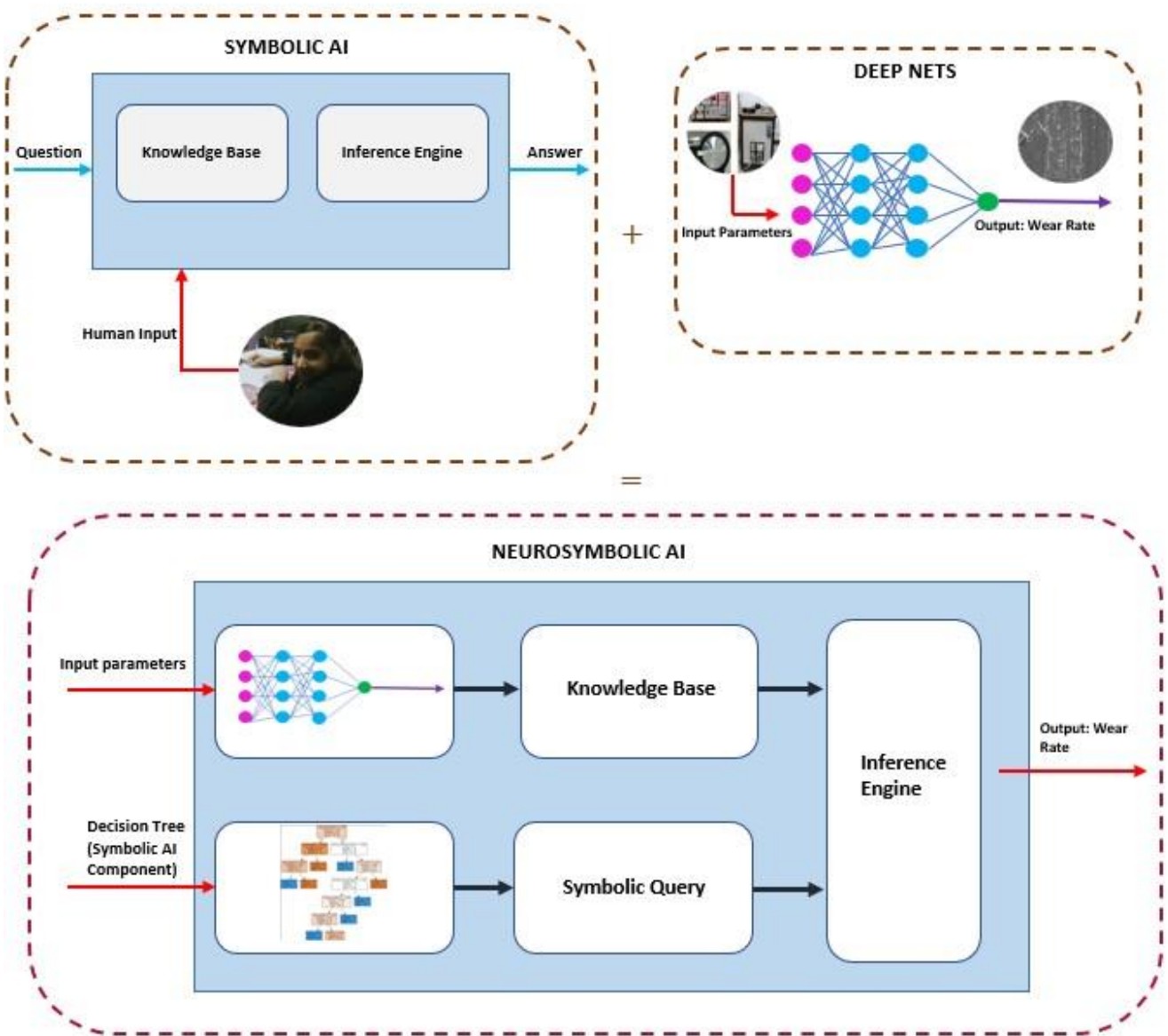

**Figure 7.** NSAI framework to predict the wear rate [32].

The fusion of these two techniques in the neurosymbolic algorithm provides a robust forecasting tool for wear rates in Al/SiC MMCs. The combination allows the model to handle both explicit, structured relationships, and implicit, non-linear relationships in the data, thereby offering a powerful balance of interpretability and predictive accuracy. This unique blend allows for a more nuanced understanding of the effects of varying SiC weight percentages and compositional transformations on the wear rates in MMCs.

On combining the training and validation sets, a strong model should consistently show accuracy and comparable performance. Any discrepancies between the actual and estimated values in the validation set can point to over- or underfitting in the model, which would make it difficult to generalize to new data. The graphs of the actual versus predicted values for the training and validation sets for the two models, i.e., the simple ANN and the neurosymbolic model, are shown in Figures 8 and 9.

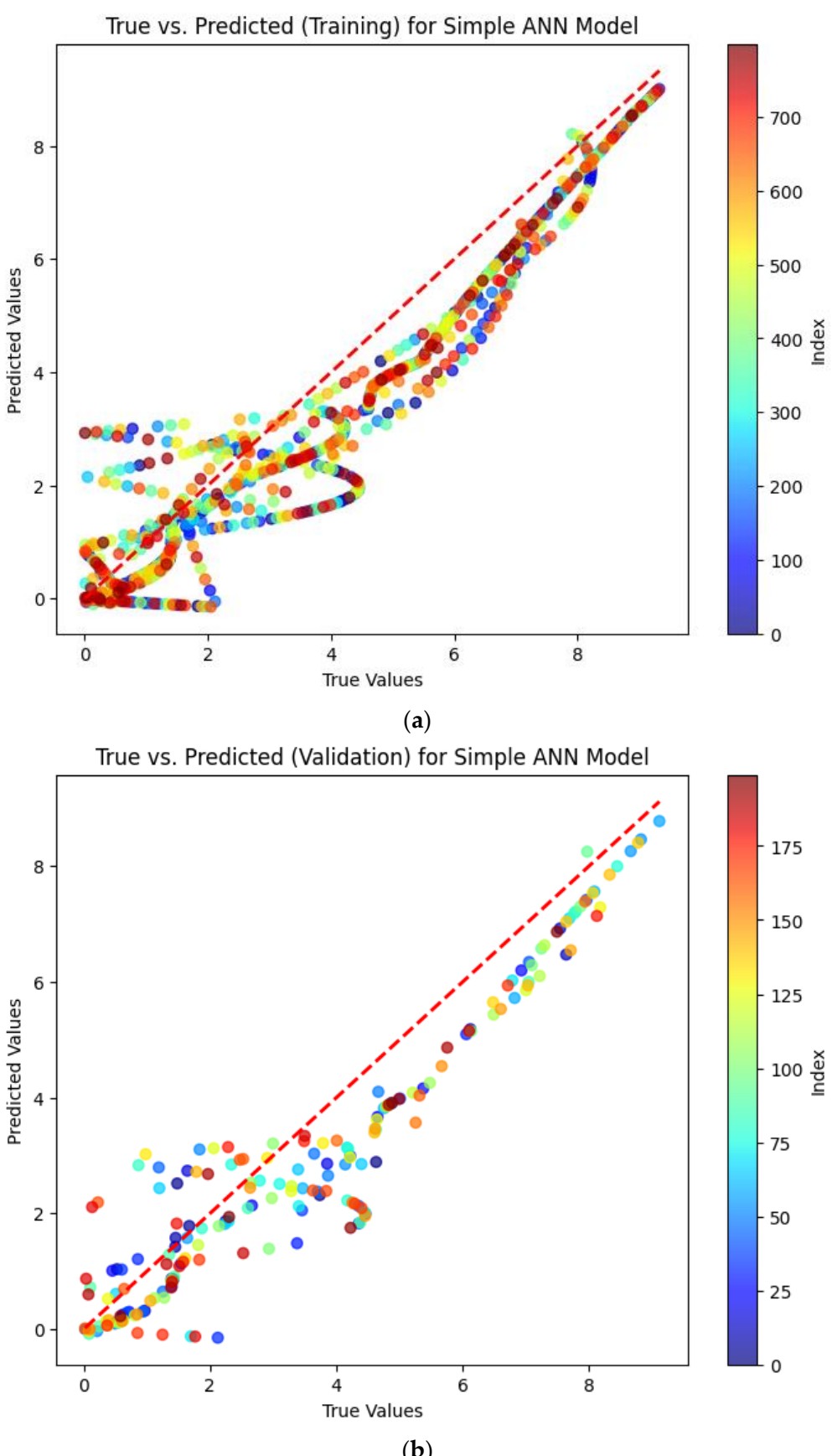

**Figure 8.** The plot obtained for simple ANN: (**a**) true vs. predicted values for training; (**b**) true vs. predicted values for validation.

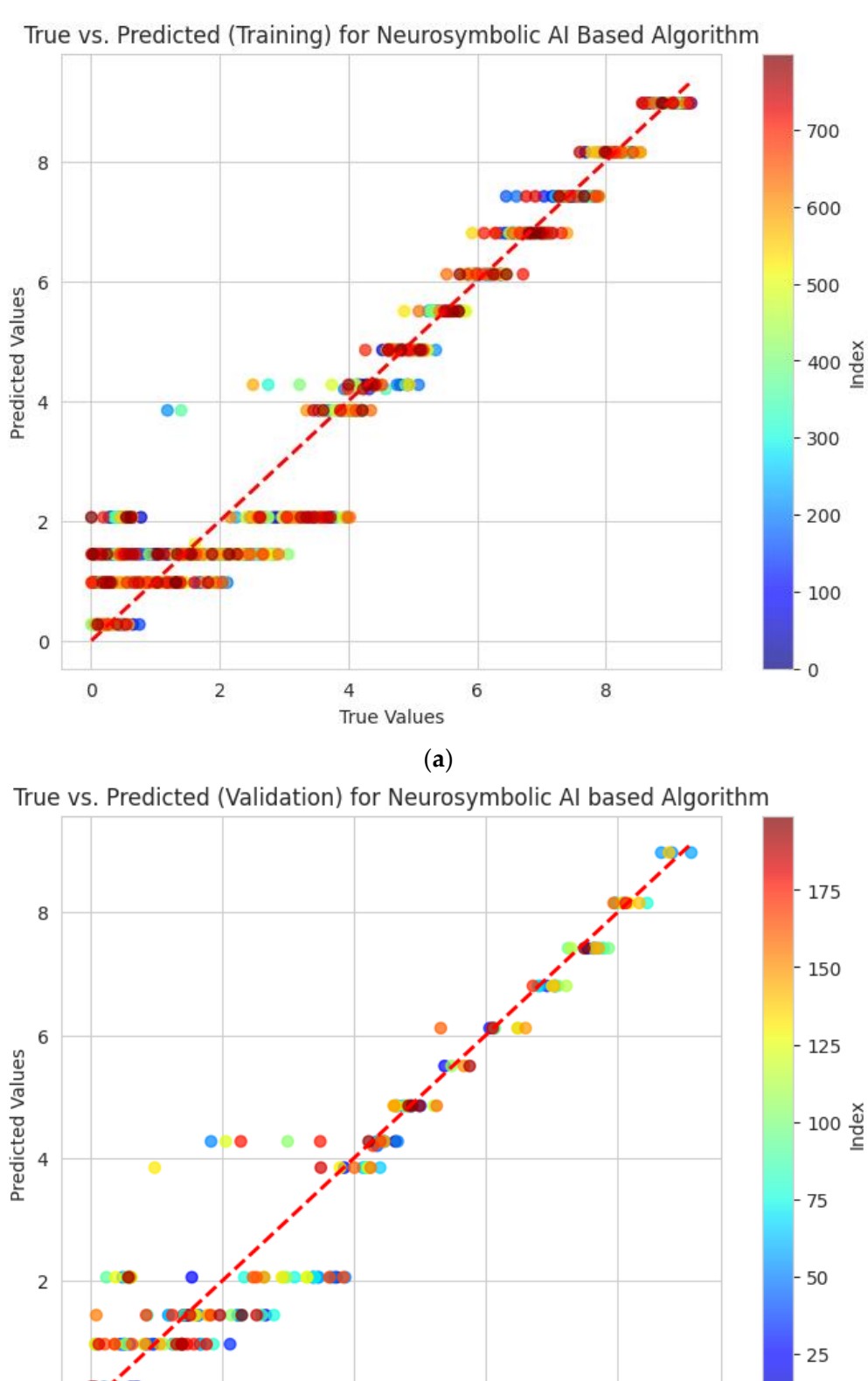

**Figure 9.** The plot obtained for neurosymbolic-based developed algorithm: (**a**) true vs. predicted values for training; (**b**) true vs. predicted values for validation.

## 5. Conclusions

In conclusion, this research investigation into predicting wear rates in Al/SiC metal matrix composites has yielded significant findings and valuable insights. Through meticulous spectroscopic analysis, we successfully determined the distribution of different material constituents within the composites containing varying weight percentages of silicon carbide (SiC) (0%, 3%, and 5%). Our findings revealed notable changes in the composition and distribution of multiple elements due to the integration of SiC.

The comparative analysis between the simple artificial neural network (ANN) and the neurosymbolic algorithms demonstrated the superior performance of the neurosymbolic approach. The neurosymbolic algorithm consistently outperformed the other model, with lower mean squared error (MSE) values and higher R-squared ($R^2$) values during both the training and validation phases. These results underscore the potential of neurosymbolic programming to generate more precise and resilient predictions, particularly for predicting wear rates in Al/SiC metal matrix composites.

While the outcomes of this study are encouraging, it is essential to acknowledge that the model's performance may vary based on specific material attributes and operational conditions. Thus, future research should focus on validating and expanding these findings across a broader range of materials and conditions. This comprehensive investigation will provide deeper insights into the wear behavior of Al/SiC metal matrix composites and facilitate the development of more robust and efficient materials for diverse engineering applications.

**Author Contributions:** Conceptualization, V.S.J. and A.M.; methodology, V.S.J.; software, A.M.; validation, A.M. and V.S.J.; formal analysis, A.M.; investigation, V.S.J. and A.M.; resources, V.S.J.; data curation, A.M.; writing—original draft preparation, A.M.; writing—review and editing, V.S.J.; visualization, A.M.; supervision, V.S.J.; project administration, V.S.J. All authors have read and agreed to the published version of the manuscript.

**Funding:** This research received no external funding.

**Data Availability Statement:** Data are available upon request by readers.

**Conflicts of Interest:** The authors declare no conflict of interest.

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
