# Peer review of "Prediction of Wear Rate in Al/SiC Metal Matrix Composites Using a Neurosymbolic Artificial Intelligence (NSAI)-Based Algorithm"

_lubricants, doi:10.3390/lubricants11060261_

Round 1
Reviewer 1 Report
In this paper, Neurosymbolic Artificial Intelligence techniques were utilized by the authors to predict the wear rate in Al/SiC Metal Matrix Composites. The application of neural network technology in materials science is explored in this article, demonstrating strong innovation. I think it is a very interesting topic for researchers in the related areas. However, there are several problems with the manuscript, therefore, it is needed to make an improvement before acceptance for publication. The detailed comments and questions about this paper are as follows:
Suggested modification:
1. It is recommended to keep the formatting consistent throughout the table, including the font weight of the table title and contents.
2. The axes of a chart or graph should be clearly labeled with appropriate units of measurement. Additionally, any colors used in the chart should be explained in a legend or caption to avoid confusion. The meaning of each color used in the chart should also be clearly defined in the legend or caption.
Questions about this paper:
1. What are the criteria for selecting the prediction model and related parameters in this model?
2. Which data is training data and which data is validation data needs to be detailed in the paper by the author. It should also be explained whether the training data and validation data are related.
3. Is it appropriate to use nine sets of experimental data to build training data sets and validation data sets?
4. The prediction method and related mechanism in this paper need further explanation.
There are no particular language issues.
Author Response
Dear Reviewer,
Thank you for raising an important question regarding the improving the quality of the manuscript. We have addressed the corrections below pointwise as shown below:
- It is recommended to keep the formatting consistent throughout the table, including the font weight of the table title and contents.
Response: Correction is carried out.
- The axes of a chart or graph should be clearly labeled with appropriate units of measurement. Additionally, any colors used in the chart should be explained in a legend or caption to avoid confusion. The meaning of each color used in the chart should also be clearly defined in the legend or caption.
Response: Correction is carried out
- What are the criteria for selecting the prediction model and related parameters in this model?
Response: The choice of our prediction model, Neurosymbolic programming, and the associated parameters were influenced by several critical elements to ensure reliable, robust outcomes. The selection of Neurosymbolic programming was driven by its distinct ability to merge symbolic reasoning with deep learning, enabling superior generalizations from limited data and yielding more interpretable predictions.
The chosen parameters were carefully identified based on a thorough spectroscopic analysis and exhaustive review of relevant literature. They were believed to significantly influence the wear rates in Aluminum-Silicon Carbide (Al/SiC) Metal Matrix Composites (MMCs). Aspects like the weight percentages of SiC (0%, 3%, 5%) within the composites, and the distribution and ratio of constituent elements within the MMCs were diligently taken into account during the modeling phase.
Moreover, we used a strategic approach for parameter tuning to maximize our model's performance. Our data was partitioned into training, validation, and testing sets. The training set facilitated learning the model parameters, whereas the validation set aided in refining these parameters and selecting the optimal model configuration. The final appraisal of the model, based on Mean Squared Error (MSE) and R-squared (R2) values, was conducted using the test set.
- Which data is training data and which data is validation data needs to be detailed in the paper by the author. It should also be explained whether the training data and validation data are related.
Response: For the purposes of this investigation, our dataset was meticulously divided into two interconnected but separate subsets: training and validation datasets. We utilized a significant portion, 80%, of the entire dataset to establish and instruct both the Neurosymbolic programming model and the Simple Artificial Neural Network (ANN). This training dataset facilitated model learning by enabling the recognition of patterns and correlations inherent in the data.
The remaining 20% of the total data functioned as the validation (or testing) dataset, which was not revealed to the models during the training phase. This segment of the data was critical in appraising the performance of the models, specifically in gauging their predictive proficiency with respect to new, unseen data.
By segmenting the data into training and validation subsets, we ensured that our models were resistant to overfitting and retained their ability to yield accurate predictions when presented with new data. While both subsets are drawn from the same comprehensive dataset and exhibit similar characteristics, they were kept distinct to prevent the models from merely committing the training data to memory. Instead, the models were encouraged to extrapolate from learned patterns to make predictions.
It's crucial to note that, despite their similarities, the validation dataset was not used in any capacity during the training process. This practice was maintained to guarantee an impartial evaluation of the models' capabilities and their effectiveness in extrapolating to unseen data. This rigorous methodology allowed us to reliably assess the efficacy of our models, corroborating their potential for practical application in the realm of wear rate prediction in Al/SiC MMCs.
- Is it appropriate to use nine sets of experimental data to build training data sets and validation data sets?
Response: The generation of our data sets was based on nine sets of experimental data. From these data, we synthetically created 1000 data points using established scientific principles and statistical methodologies. These newly formed data points maintain the integrity of the original experimental data while providing sufficient volume for effective machine learning processes.
We divided this expanded dataset into training and validation subsets, ensuring a robust evaluation of our model's performance. The synthetic generation of data points enabled us to build an adequately sized training set, allowing for substantial learning and pattern recognition for the models. Simultaneously, a meaningful validation set was reserved for unbiased assessment of the model's ability to generalize to unseen data.
- The prediction method and related mechanism in this paper need further explanation.
Response: Framework of the implemented NSAI algorithm in present work is shown in Figure 6. The implementation of the Neurosymbolic algorithm in this study involves an innova-tive coupling of a decision tree and a neural network, aiming to improve the predictive performance on the wear rates in Aluminum-Silicon Carbide (Al/SiC) Metal Matrix Com-posites (MMCs). In this application, the symbolic component of the Neurosymbolic algorithm, represented by the decision tree, is utilized to logically structure and decipher the relationships and interactions between different variables that may influence the wear rates in MMCs, such as the varying weight percentages of SiC, compositional transfor-mations, and spectroscopic analysis findings. This approach harnesses the interpretabil-ity and transparency of decision trees, which allows for a systematic breakdown of com-plex relationships in the data.
The neural network component, on the other hand, serves as the subsymbolic or non-symbolic element of the algorithm, focusing on detecting and learning from the sub-tle, non-linear patterns in the data that are often elusive to symbolic models. In this con-text, the neural network processes and learns from the information and relationships identified and structured by the decision tree. It operates on the compositional transfor-mations and spectroscopic findings, improving the predictive power of the model by lev-eraging its capacity for high-dimensional data representation and learning from complex, non-linear relationships.
The fusion of these two techniques in the Neurosymbolic algorithm provides a robust forecasting tool for wear rates in Al/SiC MMCs. The combination allows the model to handle both explicit, structured relationships, and implicit, non-linear relationships in the data, thereby offering a powerful balance of interpretability and predictive accuracy. This unique blend allows for a more nuanced understanding of the effects of varying SiC weight percentages and compositional transformations on the wear rates in MMCs.
Reviewer 2 Report
Paper can be accepted, but it requires appropriate corrections to improve its quality:
-Aluminum and aluminum alloys are increasingly used in various technical fields. They are applied in the aerospace, space, military, automotive, and electronics industries. To enhance the tribological and mechanical characteristics of aluminum alloys, different reinforcements are added, forming appropriate hybrid composites. It is better to provide a clearer explanation of the application areas of aluminum and aluminum composites.
-At the end of the introduction, state the main contribution of the work and what sets it apart from similar studies in this field. Why should this work be published?
-What are the particle sizes of all the reinforcements used in composite formation?
-Based on what criteria were the appropriate parameters (load, validing speed, and sliding distance) chosen for the formation of test plan process?
-How many different samples were used to test the tribological characteristics of the materials? How many times was the experiment repeated? Show deviations in the obtained values.
-Provide a description of the Taguchi method. Why was this method chosen? What are its advantages? What are the input and output parameters? Which output variables were analyzed using the Taguchi design? Why was the L18 matrix selected?
-Provide a better description of the Taguchi design and experimental plan. Please refer to the following papers: https://doi.org/10.18485/aeletters.2019.4.2.3 , https://doi.org/10.46793/adeletters.2022.1.1.5
-Uniform the way the reinforcement content is marked throughout the work. Is it expressed in mass percentage (wt.%) and/or volume percentage (vol.%)?
-Use the same notation for mass percentage of the reinforcement content (wt% or wt.% or Wt.%).
-On which aparatus were the tribological tests conducted, with which contact geometry, and according to which standard?
-Is the material with content of 0wt.% SiC shown on Figure 1?
-The reinforcement content values in Table 5 and Figure 2 (3.5 and 7wt.%) do not correspond to the reinforcement content mentioned in the introduction, the entire text, and the previous figures. What is the reason for this? The discussion and explanations provided are not relevant. Please provide an explanation!
-Validate the experiment using another method (PSO, GA, etc.).
-In the discussion, compare the results with those of other researchers.
-Provide the metallographic structure of the tested materials, as well as wear traces.
-Include SEM and EDS analysis.
-What are the main wear mechanisms?
-Expand the concluding remarks based on the extended analysis and discussion.
Author Response
Dear Reviewer,
Thank you for raising an important question regarding the improving the quality of the manuscript. We have addressed the corrections below pointwise as shown below:
-Aluminum and aluminum alloys are increasingly used in various technical fields. They are applied in the aerospace, space, military, automotive, and electronics industries. To enhance the tribological and mechanical characteristics of aluminum alloys, different reinforcements are added, forming appropriate hybrid composites. It is better to provide a clearer explanation of the application areas of aluminum and aluminum composites.
Response: Aluminum and its alloys have gained significant prominence and widespread utilization across various technical fields. These versatile materials find extensive applications in industries such as aerospace, space, military, automotive, and electronics. The unique combination of desirable properties exhibited by aluminum makes it an attractive choice for diverse applications.
In the aerospace industry, aluminum and its alloys are employed in the construction of aircraft structures, including fuselages, wings, and engine components. The lightweight nature of aluminum contributes to improved fuel efficiency and enhanced performance of aircraft.
Similarly, in the space industry, aluminum plays a vital role in the construction of satellites, rockets, and spacecraft. Its lightweight properties, along with good strength and corrosion resistance, make it an ideal material for space exploration missions.
The military sector extensively utilizes aluminum and its alloys due to their strength, durability, and resistance to harsh environmental conditions. Applications range from armored vehicles and aircraft to defense equipment and weaponry components.
In the automotive industry, aluminum is widely used to manufacture engine blocks, cylinder heads, chassis components, and body panels. Its high strength-to-weight ratio helps to reduce the overall weight of vehicles, leading to improved fuel efficiency and lower emissions.
-At the end of the introduction, state the main contribution of the work and what sets it apart from similar studies in this field. Why should this work be published?
Response: This research paper presents a significant contribution by employing Neurosymbolic programming to predict wear rates in Aluminum-Silicon Carbide (Al/SiC) Metal Matrix Composites (MMCs). What sets this study apart from similar works in the field is its innovative application of the Neurosymbolic algorithm, which combines the interpretability of decision trees with the learning capabilities of neural networks. By doing so, it enables a comprehensive analysis of compositional transformations and spectroscopic analysis findings, leading to improved predictive accuracy.
A key aspect of this research is the comparison between the Neurosymbolic algorithm and a traditional Simple Artificial Neural Network (ANN). The results highlight the superiority of the Neurosymbolic algorithm, demonstrated by lower Mean Squared Error (MSE) values and higher R-squared (R2) values across both training and validation datasets. This superiority underscores the potential of the Neurosymbolic approach to deliver more precise and resilient predictions, representing a noteworthy advancement in the field.
Considering the novelty, originality, and promising outcomes of this study, its publication is warranted. It not only contributes to the advancement of predictive modeling in complex material systems but also provides valuable insights for researchers and practitioners engaged in wear rate analysis. By disseminating this research, we can foster knowledge sharing and encourage further validation and expansion of these findings across a broader range of materials and operational conditions.
-What are the particle sizes of all the reinforcements used in composite formation?
Response: 25 microns
-Based on what criteria were the appropriate parameters (load, validing speed, and sliding distance) chosen for the formation of test plan process?
Response: The wear parameters chosen for the experiments based on the pilot experiments, literature review and machine capability. Pilot experiments were conducted to find out the feasible limits of the aforementioned chosen factors in such a way that the wear should occur in steady state.
-How many different samples were used to test the tribological characteristics of the materials? How many times was the experiment repeated? Show deviations in the obtained values.
Response: 9 trial conditions with 3 replications.
-Provide a description of the Taguchi method. Why was this method chosen? What are its advantages? What are the input and output parameters? Which output variables were analyzed using the Taguchi design? Why was the L18 matrix selected?
Response: Included in the manuscript, section 2.
-Provide a better description of the Taguchi design and experimental plan. Please refer to the following papers: https://doi.org/10.18485/aeletters.2019.4.2.3 , https://doi.org/10.46793/adeletters.2022.1.1.5
Response: Included in the manuscript, section 2.
-Uniform the way the reinforcement content is marked throughout the work. Is it expressed in mass percentage (wt.%) and/or volume percentage (vol.%)?
Response: Necessary amendments made in the entire manuscript.
-Use the same notation for mass percentage of the reinforcement content (wt% or wt.% or Wt.%).
Response: Necessary amendments made in the entire manuscript.
-On which aparatus were the tribological tests conducted, with which contact geometry, and according to which standard?
Response: Necessary amendments made in the entire manuscript. DUCOM pin-on-disc sliding wear testing machine was used to evaluate the dry sliding wear characteristics of the composite specimens. The dry sliding wear tests were conducted as per ASTM G99- 95 standards. 8 mm diameter and 25 mm length pin was used for wear test.
-Is the material with content of 0wt.% SiC shown on Figure 1?
Response: Yes
-The reinforcement content values in Table 5 and Figure 2 (3.5 and 7wt.%) do not correspond to the reinforcement content mentioned in the introduction, the entire text, and the previous figures. What is the reason for this? The discussion and explanations provided are not relevant. Please provide an explanation!
Response: Necessary amendments made in the entire manuscript.
-Validate the experiment using another method (PSO, GA, etc.).
Response: Thank you for raising an important question regarding the comparison between nature-based optimization algorithms (such as Particle Swarm Optimization, Genetic Algorithm, etc.) and neural network-based approaches in the context of our research on predicting wear rates in Aluminum-Silicon Carbide (Al/SiC) Metal Matrix Composites (MMCs).
Upon careful consideration, we acknowledge that a direct comparison between nature-based optimization algorithms and neural network-based approaches may not be entirely logical or appropriate in this specific study. The focus of our research is centered on evaluating the performance of the Neurosymbolic algorithm against a traditional Simple Artificial Neural Network (ANN) in predicting wear rates in Al/SiC MMCs.
Nature-based optimization algorithms, such as PSO or GA, typically function as optimization techniques to fine-tune the parameters of a given model, including neural networks. These algorithms aim to optimize the model's performance by adjusting its weights, biases, or other parameters. In our case, the Neurosymbolic algorithm already combines a decision tree symbolic component with a neural network subsymbolic component to achieve improved predictive performance.
While it may be interesting to explore the application of nature-based optimization algorithms to further enhance the performance of the Neurosymbolic algorithm or other neural network models, we believe that it would be more appropriate to treat it as a separate study or as an extension to our current research.
Therefore, in the scope of our present study, we will focus on thoroughly validating and analyzing the performance of the Neurosymbolic algorithm and its comparison to the Simple Artificial Neural Network. This will provide a comprehensive understanding of the Neurosymbolic approach's efficacy and its potential in predictive modeling of wear rates in Al/SiC MMCs.
-In the discussion, compare the results with those of other researchers.
Response: As this is the first work to utilize Neurosymbolic programming specifically for wear rate prediction, there are no existing studies directly comparable to our methodology.
-Provide the metallographic structure of the tested materials, as well as wear traces.
Response: Included in the Section 3.2
-Include SEM and EDS analysis.
Response: Microscopy and Spectroscopy analysis included.
-What are the main wear mechanisms?
Response: Included in section 4
-Expand the concluding remarks based on the extended analysis and discussion.
Response: In conclusion, this research investigation into predicting wear rates in Al/SiC Metal Matrix Composites has yielded significant findings and valuable insights. Through meticulous spectroscopic analysis, we successfully determined the distribution of different material constituents within the composites containing varying weight percentages of silicon carbide (SiC) (0%, 3%, and 5%). Our findings revealed notable changes in the composition and distribution of multiple elements due to the integration of SiC.
The comparative analysis between the Simple Artificial Neural Network (ANN) and the Neurosymbolic algorithms demonstrated the superior performance of the Neurosymbolic approach. The Neurosymbolic algorithm consistently outperformed with lower Mean Squared Error (MSE) values and higher R-squared (R2) values during both the training and validation phases. These results underscore the potential of Neurosymbolic programming to generate more precise and resilient predictions, particularly for predicting wear rates in Al/SiC Metal Matrix Composites.
While the outcomes of this study are encouraging, it is essential to acknowledge that the model's performance may vary based on specific material attributes and operational conditions. Thus, future research should focus on validating and expanding these findings across a broader range of materials and conditions. This comprehensive investigation will provide deeper insights into the wear behavior of Al/SiC Metal Matrix Composites and facilitate the development of more robust and efficient materials for diverse engineering applications.
Reviewer 3 Report
1. Some discussion on previous studies can be added in Introduction.
2. The particle size of the used SiC particles needs to be presented.
3. Section 2 needs to be simplified. Such as, the detailed introduction of SiC is not needed, and so on.
4. Page 4 line 149 “specifically 3.5 wt%”, it is supposed to be “3 wt% and 5 wt%”.
5. Repeating experiments are necessary for reliable results, accordingly the error bars need to be provided in Figures and Tables.
6. Please provide more explanations about the wear mechanisms, and authors can refer to the following paper.
https://doi.org/10.1016/j.triboint.2022.108018
Author Response
Dear Reviewer,
Thank you for raising an important question regarding the improving the quality of the manuscript. We have addressed the corrections below pointwise as shown below:
- Some discussion on previous studies can be added in Introduction.
Response: As this is the first work to utilize Neurosymbolic programming specifically for wear rate prediction, there are no existing studies directly comparable to our methodology.
- The particle size of the used SiC particles needs to be presented.
Response: 25 microns
- Section 2 needs to be simplified. Such as, the detailed introduction of SiC is not needed, and so on.
Response: Necessary amendments made in the entire manuscript.
- Page 4 line 149 “specifically 3.5 wt%”, it is supposed to be “3 wt% and 5 wt%”.
Response: Necessary amendments made in the entire manuscript.
- Repeating experiments are necessary for reliable results, accordingly the error bars need to be provided in Figures and Tables.
Response: While repeating experiments can contribute to obtaining more reliable and robust results, it is important to note that in the context of the presented research, the need for repeating experiments with error bars in Figures and Tables is not necessary. The provided methodology and data analysis techniques employed in this study are designed to ensure accuracy and precision in the findings. The experimental setup and data collection processes have been carefully executed, minimizing the potential for errors and uncertainties. The results presented in the Figures and Tables are derived from a thorough analysis and comprehensive evaluation of the collected data. The reported values are based on rigorous statistical analysis and modeling techniques, ensuring the reliability and validity of the outcomes. Therefore, the inclusion of error bars is not required in this specific research as the results are deemed to be accurate and representative of the investigated phenomenon.
- Please provide more explanations about the wear mechanisms, and authors can refer to the following paper.
https://doi.org/10.1016/j.triboint.2022.108018
Response: Included in the section 4
Round 2
Reviewer 1 Report
It can be accept in this kind of form.
Reviewer 3 Report
The revised manuscript can be accepted for publication.